**Subject Category:**
Biology (whole organism)

ecology/evolution/genetics

QTL analysis, *Heliothis virescens*, sex pheromone, delta-11-desaturase, point mutation

**Author for correspondence:**
Astrid T. Groot
e-mail: a.t.groot@uva.nl

†Present address: DKMS Life Science Lab GmbH, Blasewitzer Str. 43, 01307 Dresden, Germany

# Within-population variability in a moth sex pheromone blend, part 2: selection towards fixation

Astrid T. Groot[1,2], Michiel van Wijk[1],
Ernesto Villacis-Perez[1], Peter Kuperus[1],
Gerhard Schöfl[2,†], Dennis van Veldhuizen[1]
and David G. Heckel[2]

[1]Institute for Biodiversity and Ecosystem Dynamics, University of Amsterdam, Science Park 904, 1098 XH Amsterdam, The Netherlands
[2]Department Entomology, Max Planck Institute for Chemical Ecology, Hans Knoell Strasse 8, 07745 Jena, Germany

ATG, 0000-0001-9595-0161

To understand how variation in sexual communication systems evolves, the genetic architecture underlying sexual signals and responses needs to be identified. Especially in animals where mating signals are important for mate recognition, and signals and responses are governed by independently assorting genes, it is difficult to envision how signals and preferences can (co)evolve. Moths are a prime example of such animals. In the noctuid moth *Heliothis virescens*, we found within-population variation in the female pheromone. In previous selection experiments followed by quantitative trait locus (QTL) analysis and expression analysis of candidate desaturase genes, we developed a model involving a *trans*-acting repressor of the delta-11-desaturase. In our current study with new selection lines, we fixed the most extreme phenotype and found a single underlying mutation: a premature stop codon in the first coding exon of delta-11-desaturase, which we could trace back to its origin in the laboratory. Interestingly, we found no pleiotropic effects of this knock-out mutation on the male physiological or behavioural response, or on growth or fertility. This finding is in contrast to *Drosophila melanogaster*, where a single desaturase gene affects both female pheromone production and male behavioural response, but similar to other Lepidoptera where these traits are under independent genetic control. To our knowledge, this is the first time that a single point mutation has been identified that underlies the phenotypic variation in the pheromone signal of a moth.

# 1. Introduction

Finding the right mating partner is one of the most important goals in the life of sexually reproducing animals, and many animals use sex pheromones to attract potential mates [1]. Even though mate choice behaviour has been studied extensively in many animal species, little is known about the genes underlying sexual signals or responses to these signals [2,3]. Identification of these genes is necessary to understand how variation in sexual communication systems has evolved and contributes to speciation [4].

The genetic basis of sexual communication has mostly been studied in model systems such as fruit flies [5–8], crickets [9–11], fish [12–14] and moths [15–20]. However, only a few studies have actually identified genes underlying phenotypic variation in mate signals or responses [7,10,17,18,21,22], which has limited progress in assessing the role of sexual selection in diversification and speciation [4].

In some animals, both sexual communication signals and responses are determined by the same genes or at least the same genomic locations [7,11,23–25]. The most striking example of a single gene affecting both functions is *desat1* in *Drosophila melanogaster* [6,21]. Different transcripts, each encoding the same desaturase enzyme, are expressed in neural and non-neural tissues, and transcript-specific RNAi of *desat1* demonstrated effects on both pheromone emission and perception. Such genetic coupling has not been found in moth sexual communication systems [17,18,26–30], which makes it difficult to envision how signals and preferences can (co)evolve. Especially when these mating signals are mostly important for mate recognition, stabilizing selection would seem to impede diversification [31–33].

In moths, sexual communication has traditionally been viewed as a fine-tuned system important for mate recognition that reinforces interspecific reproductive isolation [32]. The female signal is exclusively chemical, produced in a well-defined gland and readily quantified. The male behavioural response is robust and specific, and the well-studied pheromone sensory system serves as an important model for decoding olfactory preference in general. However, even though the sex pheromones of more than 1600 moth species have been identified (Pherobase.com), the mechanisms of evolutionary change in pheromone systems are still obscure, because the genetic basis of these changes has been identified in only a few cases [15–18,20].

Female moth pheromones consist of a mixture of modified fatty acids of various lengths, which may have one or more double bonds in the hydrocarbon chain and a terminal alcohol, aldehyde or acetate ester functional group (reviewed by Jurenka and co-workers [34–36]). Enzymes which catalyse modifications include desaturases, which introduce one or more double bonds, and fatty-acyl reductases, which convert the CoA thioesters to alcohols. Alcohols themselves can be active pheromone components, or they may be converted to aldehydes through alcohol oxidase (aldehydes can be converted to alcohols through aldehyde reductase). Alcohols can also be converted to acetates (i.e. acetate esters). All involved enzymes are members of large gene families and the challenge has been to identify the ones specifically responsible for determining the species-specific pheromone blend. To our knowledge, *Ostrinia nubilalis* is so far the only species for which the gene responsible for the two pheromone races has been identified [18].

Previously, we found that natural populations of the noctuid moth *Heliothis virescens* show within-population variation in female pheromones and that this variation responds to artificial selection [37]. The essential sex pheromone components of *H. virescens* females are (Z)-11-hexadecenal (Z11−16:Ald) and (Z)-9-tetradecenal (Z9−14:Ald) which are both necessary to attract conspecific males [38–40]. Additional compounds in the pheromone gland are tetradecanal (14:Ald), hexadecanal (16:Ald), (Z)-7-hexadecenal (Z7−16:Ald), (Z)-9-hexadecenal (Z9−16:Ald) and (Z)-11-hexadecen-1-ol (Z11-16:OH) [38–40]. We have recently shown that all these compounds in the pheromone gland are also emitted [41]. Extensive sampling of natural populations revealed a substantial proportion of females (so-called High females) with much higher ratios of 16:Ald/Z11−16:Ald than had previously been characterized [37]. By selecting females based on this ratio, we constructed High- and Low-ratio selection lines with which we conducted quantitative trait locus (QTL) analysis. We found one QTL explaining 11−21% of the phenotypic variance in the ratio of two pheromone compounds [37]. Our main candidate gene was LPAQ delta-11-desaturase, because the two compounds constituting the selected ratio differ only by the presence or absence of a double bond at the 11th position, and qPCR results showed significantly higher expression levels of this and not other desaturases in the females having more of the unsaturated compounds [37]. However, because the QTL mapped to a different genomic location than the LPAQ desaturase, a model with a *trans*-acting repressor of this desaturase was invoked, which explained many features of the data.

Here we describe our approaches to identify the exact genetic change that underlies within-population polymorphism in the sex pheromone signal of *H. virescens*. We started new selection lines, this time with the even stronger selection, leading to fixation of the most extreme phenotype. With the new selection

lines, we conducted QTL analysis to assess whether we could find the same or different genomic location(s) underlying the within-population variation. Using male-informative crosses, we mapped the phenotype at a finer scale within the major QTL region. This time the main candidate gene, the LPAQ delta-11 desaturase, mapped within this region. In sequencing this gene, we found a premature stop codon in this gene that explained up to 99% of the variance in the pheromone ratio produced by females. This mutation had no effect on male responses to the female pheromone, or other pleiotropic effects. Thus, we have added another allele contributing to the within-population female pheromone variation to those that we have previously found in this species. Interestingly, unlike the case of *desat1* in *Drosophila*, this mutation has no detectable pleiotropic effect on male behavioural response.

# 2. Material and methods

## 2.1. Selection lines and heritability analysis

To reselect for the 'High' and 'Low' pheromone blends in *H. virescens*, this time we started with a laboratory population, called JEN2, which originated from larvae collected in Clayton, North Carolina in 1988 [42], reared at the Max Planck Institute for Chemical Ecology starting in 2004, subsequently supplemented with field collections from North Carolina, and reared at the University of Amsterdam since 2011. Single pair matings were randomly grouped into three groups, a 'baseline' in which no selection took place, the 'High' line, to select for females with a high ratio of 16:Ald/Z11–16:Ald (the major sex pheromone component) and the 'Low' line, to select for a low ratio (see electronic supplementary material for more details). Heritability estimates of the pheromone blends were calculated using an animal-model framework, as described in detail in [37] and further described in electronic supplementary material. After we discovered that most of the selection response was due to allele frequency changes at a single locus, we fitted a single-locus selection trajectory to the data, using specific assumptions on the genotype–phenotype relationship described in electronic supplementary material, figure S8.

## 2.2. QTL analysis

To determine the genetic basis of the selected pheromone variation in the current selection lines, we conducted QTL analysis on the separate female sex pheromone components (14:Ald, Z9–14:Ald, 16:Ald, Z7–16:Ald, Z9–16:Ald, Z11–16:Ald and Z11–16:OH). First, we hybridized and backcrossed females and males of both lines in all possible directions to determine dominance effects (electronic supplementary material).

### 2.2.1. Phenotyping

All females were phenotyped by extracting their sex pheromone glands, as described in detail in [37,43] and summarized here. Females were injected with pheromone biosynthesis activating neuropeptide (PBAN) 1–2 h before gland extraction to stimulate pheromone production. Pheromone glands were excised and extracted for 20–30 min in 50 µl hexane containing 20 ng pentadecane as internal standard, after which samples were reduced to 1–2 µl under a gentle stream of $N_2$ and injected into a splitless inlet of an HP6890 gas chromatograph coupled with a high resolution polar capillary column and a flame ionization detector (FID). Authentic standards of all the pheromone components were injected before and after each gas chromatograph sequence to assess column performance and to check the retention times of each of the components. The amount of each pheromone compound was calculated relative to the internal standard.

### 2.2.2. Genotyping

We genotyped two backcross families to the High line with the highest phenotypic variance: the female-informative backcross family 46 (BC46) (HL × H: female F1 from a High mother and Low father backcrossed to a High male) and the male-informative backcross family 35 (BC35) (H × HL), using AFLP markers and SNPs in desaturase genes (detailed in electronic supplementary material). For comparison to the backcross 6Y-R in our previous selection experiment [37], we also genotyped the female-informative family BC23, which in both cases was an LH × L backcross to the Low line. For this comparison, we mapped the candidate desaturase genes, delta-9-desaturase with signature motif

KPSE and delta-11-desaturase with signature motif LPAQ to our genetic maps (see electronic supplementary material for the primer combinations used). Delta-11-desaturase is located on the chromosome (C21) that explained most of the variance in BC46 and BC35, while delta-9-desaturase is located on the chromosome (C04) that explained most of the phenotypic variance in our previous backcross 6Y-R. We assessed whether allelic variation in these two genes was significantly correlated with the phenotypic variation in females of BC23.

### 2.2.3. Sequencing the candidate gene

As we found C21 to explain most of the pheromone variance in both BC46 and BC35, our main candidate gene was delta-11-desaturase located on this chromosome. We sequenced the coding sequence of this gene in BC35 (see electronic supplementary material for the primer combinations used) and found a premature stop codon in the CDS at 139 bp from the start codon in individuals from the High line. The segregation pattern of this stop codon was determined by designing allele-specific primers specifically on the stop codon (TAG) and the wild-type codon encoding glutamate (GAG). In this way, we were able to distinguish all three genotypes (homozygous for the stop codon SS, homozygous for the non-stop codon NN and heterozygous SN) in all backcrosses. To further differentiate the alleles of the delta-11-desaturase gene, we also sequenced the introns.

### 2.2.4. Frequency estimation of stop-codon allele from field samples

We used a Bayesian approach to estimate the frequency of the stop-codon allele in field populations ([44], equation (1) for q). This is based on a sample frequency of 0.0 for the stop-codon allele among 593 diploid individuals. We used equation (3), which is valid for a sample frequency of 0.0, to estimate the 95% credible interval.

## 2.3. Assessment of pleiotropic effects on male response

In *D. melanogaster*, the desaturase *desat1* shows a pleiotropic effect, affecting not only the adult cuticular hydrocarbon (CHC) profile that plays a role in courtship (female CHCs important for male courtship and mating, male CHCs important for reduction of male–male interactions), but also drastically altering sex pheromone perception in males [21]. Since *desat1* is the first hit when BLASTing LPAQ delta-11-desaturase in Flybase, we assessed whether the fixation of the allelic variation in delta-11-desaturase affected the male response, both electrophysiologically and behaviourally. As the allelic variation consisted of a premature stop codon in the delta-11-desaturase gene in the first coding exon, we also determined whether the effective knock-out of this gene had any pleiotropic effect on larval development and adult fertility (electronic supplementary material).

### 2.3.1. Physiological male response (EAGs)

Male electroantennagram (EAG) responses to female pheromone components were compared between the High and the Low line (specified in electronic supplementary material), as summarized here. Severed heads were placed on a glass reference probe, while a second glass electrode was mounted on the tip of the antenna. A constant air flow ($2.0 \, \text{ml s}^{-1}$) of charcoal filtered, humidified air was provided to the preparation. Stimuli were provided through an independent glass pipette which altered the airflow 5%, stimulus duration was 1 s and the inter-stimulus interval was at least 1 min to prevent adaptation. Responses to pheromone compounds were standardized using Z-3-hexen-1-ol as a reference stimulus. For the two major sex pheromone components (Z11–16:Ald and Z9–14:Ald) and their saturated counterparts (14:Ald and 16:Ald), dose–response curves were constructed. To compare EAG responses between the selection lines, linear mixed effects analysis (lme) was performed in R, using the lme4 package.

### 2.3.2. Behavioural male response (wind tunnel experiments)

To determine whether males from the two selection lines differed in their behavioural response to sex pheromone, we analysed male response in a $2 \times 1 \times 1$ m wind tunnel to a synthetic pheromone lure that we loaded with the full blend of Low females, which was shown to be attractive in the field [45]. The wind speed was $0.25 \pm 0.02 \, \text{m s}^{-1}$, and the red light was the only source of light in the room at the time of experiments. The lure was placed upwind, while males were released from a cup at the downwind side of the wind tunnel. Virgin male moths from the Low ($n = 36$) and High ($n = 20$) selection lines were

tested individually. Each male was exposed to the female pheromone plume for 3 min before opening the lid of the holding cup into the wind tunnel and allowed to fly 5 min max. The different behavioural events measured were take-off, lock-on to the source and source contact (following [46]). For the analysis, counts of males performing one of the three behavioural events were used. Behavioural responses of males in the wind tunnel were analysed as the frequency in which males responded to the pheromone plume, using a chi-squared test. The onset of these different behavioural states was compared using a Welch two-sample *t*-test. Reported *p*-values were Bonferroni-corrected.

# 3. Results

## 3.1. Selection and heritability

In the starting population, we found 2 out of 31 phenotyped females with a ratio of 16:Ald/Z11−16:Ald > 1, which we had previously characterized as High (electronic supplementary material, tables S1 and S2 and file S1). When selecting for the most extreme ratios to create a High ratio line and a Low ratio line, we started to see a response to selection in the fifth generation, when the proportion of High females increased from 5% to 17%. After this, the proportion of High females in the High line steadily increased until generation 11 (figure 1*a,b*), much faster than in our previous selection experiment (figure 1*c*). After 10 generations, the High phenotype was nearly fixed in the High line, i.e. we found greater than 90% of the phenotyped females with a pheromone ratio greater than 1 in subsequent generations (53 out of 58 phenotyped females in generation 11, 62 out of 68 phenotyped females in generation 12). The ratio greatly exceeded 1; the unsaturated compounds disappeared almost completely and were replaced with the two saturated compounds 14:Ald and 16:Ald (figure 1*a*). The total amount of all pheromone compounds was 68.6 ± 1.8 ng in the Low females ($n = 364$) and 118.6 ± 4.7 ng in the High females ($n = 160$). We measured a least-squares estimate of realized heritability in the High line overall for the ratio 16:Ald/Z11−16:Ald ($h^2 = 1.0$, figure 1*d*, while the realized heritability in the Low line was much lower ($h^2 = 0.1$, figure 1*e*).

### 3.1.1. QTL analysis

QTL analysis in both backcrosses to the High line (HL × H) (female-informative BC46 and the male-informative BC35) revealed that the variance in the ratio of 16 : Ald/Z11−16:Ald was best explained by a single QTL on chromosome 21 (figure 2). The LOD scores obtained for the QTL in both families were highly significant ($p < 0.0001$). In BC23 (LH × L), the LOD score for LPAQ delta-11-desaturase was also highly significant ($p < 0.0001$), whereas the LOD score obtained for KPSE delta-9-desaturase was not ($p = 0.158$) (figure 2*c*). Fine-scale mapping in the male-informative BC35, using a 95% Bayes credible interval, mapped the QTL onto the stop codon in LPAQ delta-11-desaturase (figure 2*d*). Specifically, the multiple imputation QTL mapping method indicated that the QTL is located at position 40 cM on chromosome 21. When we included the discovered two alleles (stop codon and wild-type) of the delta-11-desaturase gene in our QTL analysis, the location of the stop codon is mapped to position 40.7 cM on chromosome 21 (more details in the electronic supplementary material).

### 3.1.2. Comparison to our previous QTL analysis

To compare our current QTL study to our previous QTL study [37], we compared the explained variance in 16:Ald/Z11−16:Ald ratio by the QTL found previously (C04) to our current QTL (C21) (electronic supplementary material, figures S4 and S5). In contrast to our previous study, we found no evidence for a QTL on C04 in BC46 and BC35. However, in backcross family BC23, which was in the same direction as our previous backcross 6Y-R, the LOD score for C04 just missed significance at the 5% level ($p = 0.0664$) to explain the variation in Z11−16:Ald (electronic supplementary material, figure S2). Interestingly, the difference between LL and LH in BC23 was in the opposite-to-expected direction (LL females having more of the saturated compounds than LH females), similar to our previous finding in 6Y-R [37] (electronic supplementary material, figure S4).

### 3.1.3. Variation at the desaturase locus

When assessing whether the stop codon in delta-11-desaturase was present in our previous QTL map (family 6Y-R), we found the allele containing the wild-type codon encoding glutamate (GAG) in all

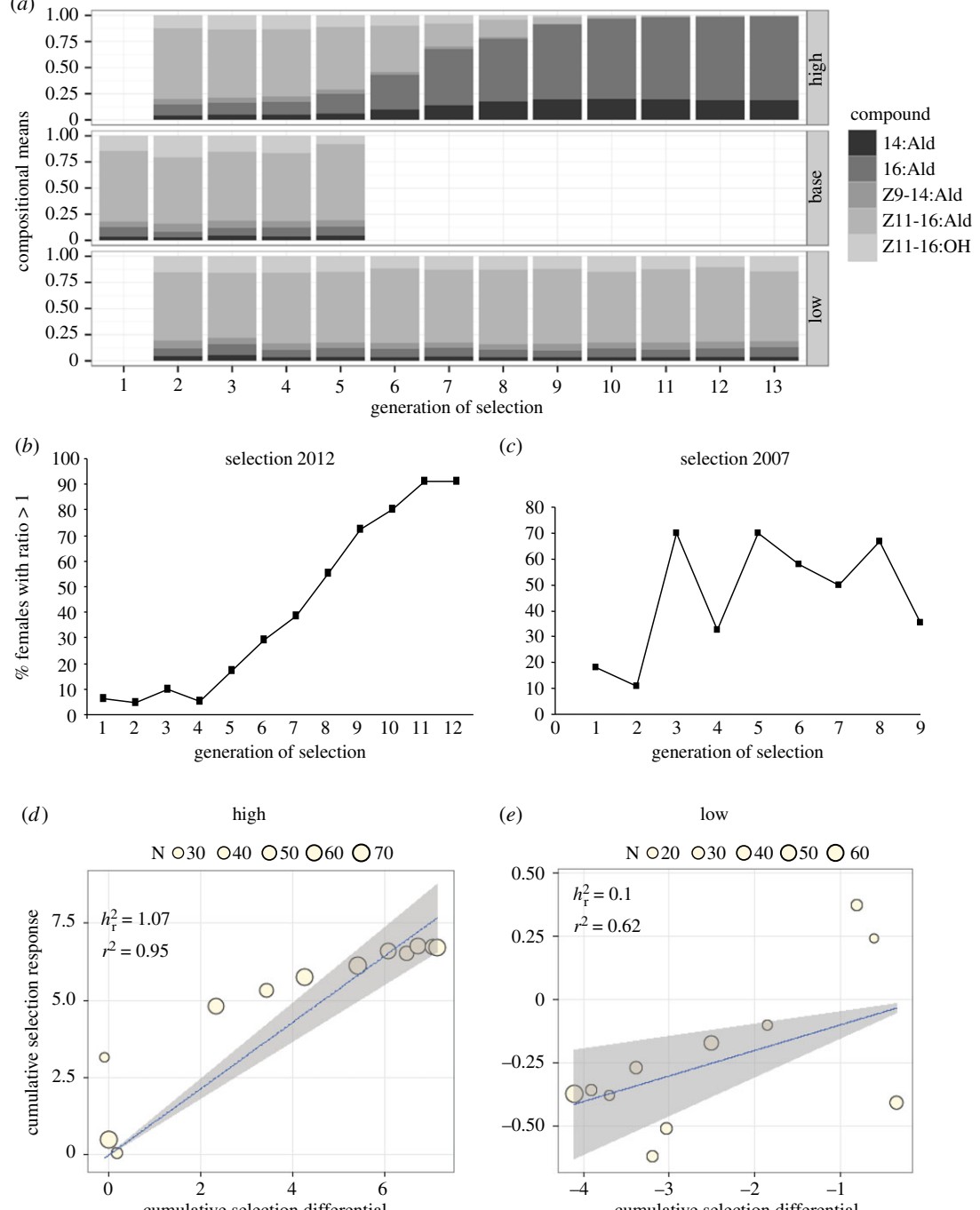

**Figure 1.** Response to selection and cumulative selection differential of the High and Low sex pheromone phenotypes. (a) Bar chart representation of the High-, Low- and base selection lines over 12 generations of selection. (b) Per cent of females in the High line with a pheromone ratio (16:Ald / Z11 − 16:Ald) greater than 1, the selection lines of 2012 in comparison to (c) the selection lines of 2007 [37]. (d) Cumulative selection differential of the High line in 2012. (e) Cumulative selection differential of the Low line in 2012.

6Y-R individuals, while none contained the stop-codon allele TAG (electronic supplementary material, file S2). We also screened a total of 593 moths collected from the field from 2005 to 2008 and in 2012 and 2013 for the presence of the stop codon, and found none. To further differentiate alleles of the delta-11-desaturase gene, we sequenced the complete gene from start to stop codon, which included two introns (electronic supplementary material figure S3 and files S3–S5). The second intron showed a size polymorphism in the previous and current backcross families and in field-collected individuals (electronic supplementary material, file S6). The sequence of the second large allele intron was identical in the selection lines, the backcross families and to a large extent identical in the

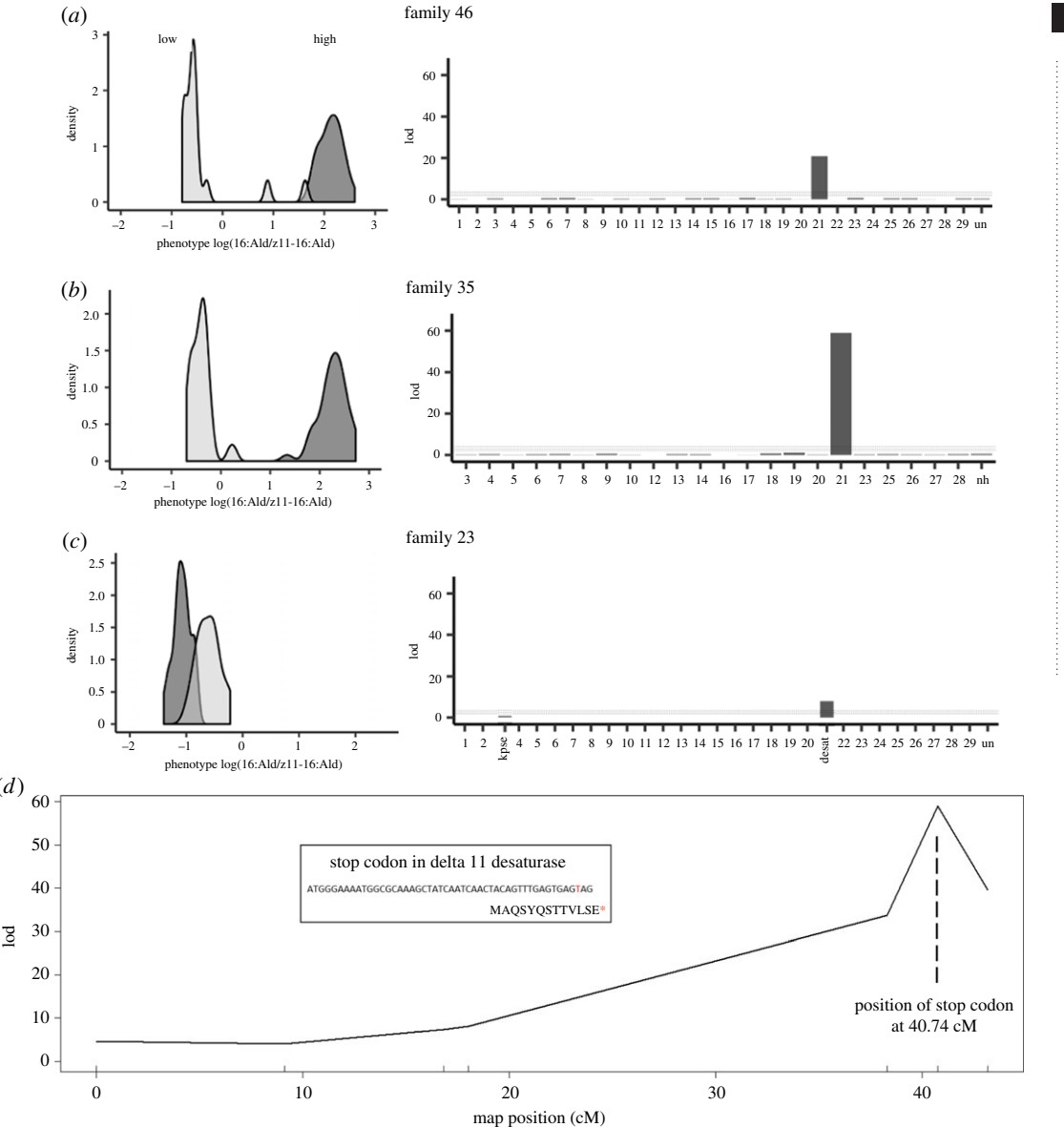

**Figure 2.** QTL analysis of three backcross families, generated from the selection lines in 2012. Left graphs show the distribution of the sex pheromone ratio 16:Ald/Z11−16:Ald in each backcross family, right graphs show the QTL analyses with LOD scores, where the dashed lines indicate LOD cut-off levels that correspond to alpha = 0.001, 0.05 and 0.2. The x-axis depicts chromosome numbers, 'un' is an artificial linkage group of unlinked markers, 'nh' is an artificial linkage group of markers from linkage groups in family 35 that could not be homologized with chromosomes in the female-informative backcross (family 46). (a) BC46: female-informative backcross HL × H ($n = 46$). Owing to the absence of crossing-over in female moths, this backcross has a resolution on the level of chromosomes. (b) BC35: male-informative backcross HL × H, used for conformation of the QTL location on chromosome 21 and for fine-scale mapping ($n = 80$). (c) BC23: female-informative backcross HL × L, which is the same direction as our previous backcross 6Y-R [37] ($n = 56$), and was only genotyped for KPSE delta-9-desaturase and LPAQ delta-11-desaturase, which correspond to chromosomes 4 and 21, respectively. LOD scores for the QTL at the delta 11 desaturase locus were always highly significant ($p < 0.0001$), whereas the LOD score at the KPSE locus was never significant. Note the difference in the scaling of the x-axis. (d) Fine-scale map of the male-informative backcross (family 35), showing the location of the stop codon in delta-11-desaturase in position 40.7 cM on chromosome 21.

field-collected females (electronic supplementary material, file S6). Since in JEN2 individuals, which was the starting population of our selection lines, we found both the large intron allele together with the stop codon, as well the small allele without the stop codon, we conclude that the stop codon mutation occurred in JEN2 in Jena, before our selection experiments. Interestingly, variation in the length of the second intron in the field-collected females was significantly correlated to the High and Low

phenotype; females with a large intron had significantly higher 16:Ald/Z11−16:Ald ratios than females with a small intron (electronic supplementary material, figure S6).

## 3.2. Tests for pleiotropic effect on male response

Owing to the point mutation resulting in a stop codon in the first coding exon, our High selection line is fixed for a non-functional delta-11-desaturase, which provided a unique opportunity to determine possible pleiotropic effects of this knock-out. We did not find any significant difference between males of the selection lines in their EAG responses to four sex pheromone components (figure 3*a*). Our wind tunnel experiments (figure 3*b*) also indicate that the presence of the stop codon in males did not affect their response to the wild-type female pheromone, as there was no significant difference between the fraction of males that took off ($\chi^2 = 0$, d.f. = 1, $p = 1$), the fraction of males that locked on after take-off ($\chi^2 = 0.5311$, d.f. = 1, $p = 0.466$) or the fraction of males that after lock-on reached the source ($\chi^2 = 0$, d.f. = 1, $p = 1$). Nor was there any difference in the time to take off ($t = -0.264$, d.f. = 50.51, $p = 1$), the time until lock-on ($t = -1.253$, d.f. = 22.969, $p = 1$) or the time until source landing ($t = -0.604$, d.f. = 17.92, $p = 1$). We also did not find a pleiotropic effect on larval growth rate or fertility (detailed in electronic supplementary material).

## 4. Discussion

The main findings of our current study are threefold: (i) we were able to select for the extremely High female sex pheromone ratio and to fix it in 12 generations, (ii) a premature stop codon which inactivates the LPAQ delta-11-desaturase accounted for most of the phenotypic variance in the selection lines, and (iii) in spite of the absence of unsaturated compounds in the female sex pheromone, we found no pleiotropic effects of the non-functional desaturase on male electrophysiological and behavioural responses, nor on larval growth rate and adult fertility.

Although we observed large responses to selection in the High lines of both our current and previous [37] experiments, they resulted in different genetic outcomes. The current experiment started with the JEN2 laboratory strain, which already contained the desaturase premature stop codon allele at a low frequency (electronic supplementary material, file S2), unknown to us at the time. After 12 generations of selection, this non-functional allele came to fixation in the High line, resulting in the near-absence of all desaturated compounds. Our previous study started with moths collected from the field in North Carolina in 2006. As the stop codon in the backcross family used in our previous study (family 6Y-R) was absent, the response we saw after eight generations in the previous experiment had a different genetic basis.

A Bayesian estimate of the frequency of the stop-codon allele in the field is 0.00084, with a 95% credible interval of [0, 0.0025] [44]. Therefore, this allele cannot account for the much higher frequency of High-ratio females found in our field samples. Moreover, the High-ratio females from the field never showed the extremely high ratio of homozygous mutant females which have virtually none of the unsaturated sex pheromone components. In sequencing the coding sequence (cds) of delta-11-desaturase of 32 individuals, we found only synonymous variation and no non-synonymous variation in the cds (electronic supplementary material, file S4). However, when checking the genomic DNA, we did find an intron-size polymorphism in the second intron after the start codon (electronic supplementary material, file S5), both in the field-collected females and in the selection lines. In fact, in the selection line, the stop codon was highly correlated with having a large second intron, which increased in frequency in our selection lines as well (electronic supplementary material, file S6). This is because the stop-codon mutation in the laboratory occurred on the background of that large second intron. Interestingly, variation in the length of the second intron in the field-collected females was significantly correlated with variation in the pheromone ratio as well. The sequences of this intron did not yield any hit in NCBI, so we currently do not have a good explanation of how this intron-size polymorphism might explain variation in the pheromone ratio.

The fact that this time we found a different QTL compared to our previous analysis is also partly due to the different types of backcrosses: in 2007, we used only a backcross family (6Y-R) that was the result of an LH × L backcross to the Low line, while this time we also analysed HL × H backcrosses to the High line. However, when we analysed the backcross (BC23) to the Low line, we also found that the stop codon in delta-11-desaturase explained most of the variance in this backcross (electronic supplementary material, figure S2). Therefore, variation at the *trans*-acting regulatory locus evident in

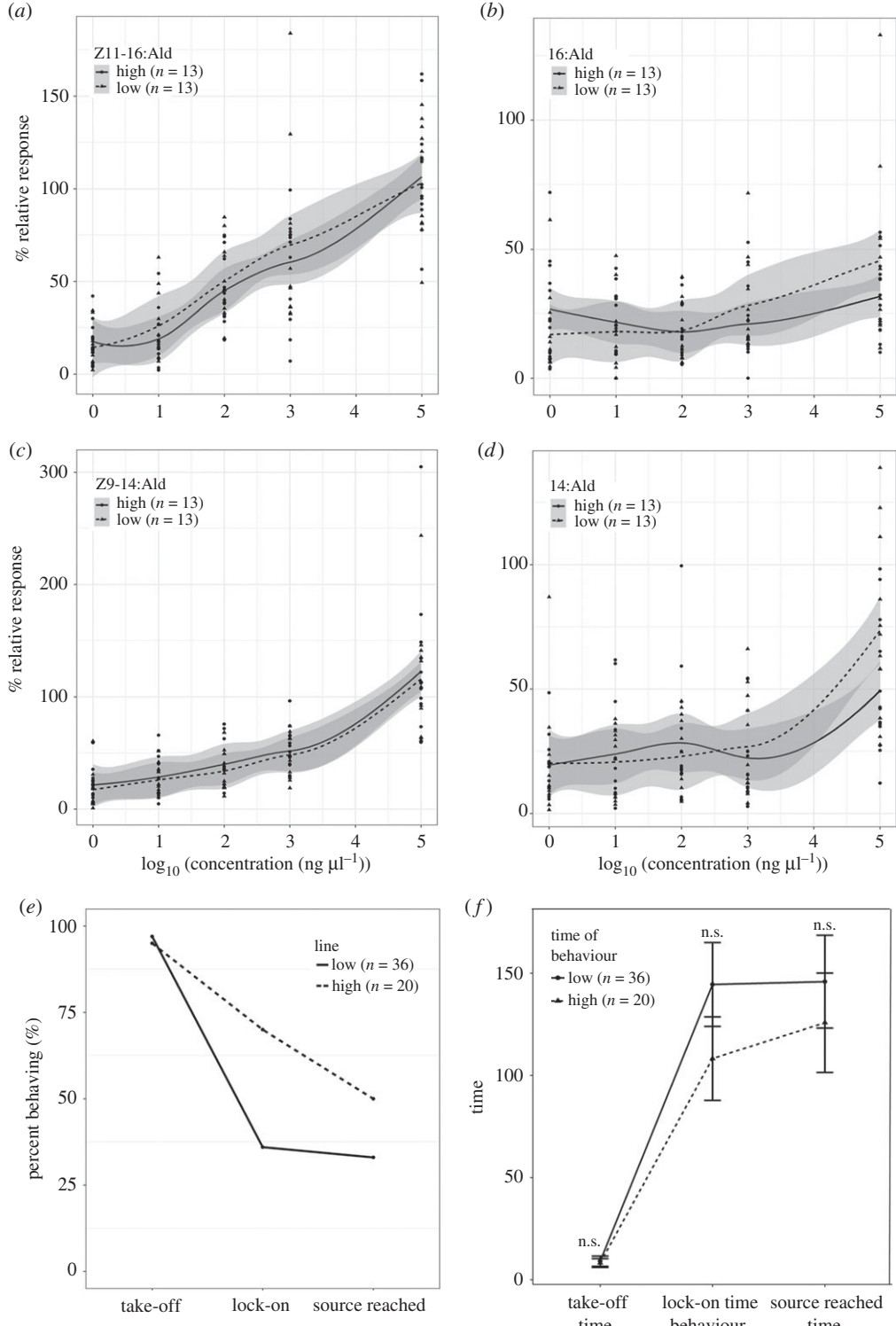

**Figure 3.** Male response of the selection lines. ($a$–$d$) Electroantennogram dose–response curves from males of the High and the Low line to four sex pheromone compounds: ($a$) the unsaturated Z11–16:Ald. ($b$) the saturated 16:Ald), ($c$) Z9–14:Ald, ($d$) 14:Ald. The $x$-axis depicts the stimulus concentration, the $y$-axis depicts the maximum relative response. Shaded areas indicate the 95% CI. ($e$,$f$) Male behavioural response in the wind tunnel. ($e$) Per cent of males exhibiting take-off, lock-on or a touch-source response. ($f$) Onset time (in s) of male take-off, lock-on and touch-source response. See text for further explanation.

the 2007 backcrosses to the Low line does not seem to have played a significant role in the current selection experiments.

There were also some notable similarities between our previous and current selection lines. In our previous as well as our current selections, there did not seem to be much of a response to selection in

the Low line, indicating that the lower limit of the ratio has already been reached, i.e. the desaturases never convert 100% of their substrates. Also, in our previous cross, we found that for linkage group C04, the High allele in a Low genomic background caused an opposite-to-expected effect, namely an apparent increase in desaturase enzyme activity resulting in a higher relative amount of Z9–14:Ald and a lower relative amount of 16:Ald than the Low allele in a Low genomic background. Interestingly, in BC23, we found a similar opposite-to-expected effect of linkage group C04 (with a marginal significance of $p = 0.066$) (electronic supplementary material, figure S2). Since we previously found that the expression level of delta-9-desaturase did not differ between High and Low females, but the expression level of delta-11-desaturase did, we postulated that a *trans*-acting repressor on Bmori_chr.23 (on which delta-9-desaturase resides) affects the expression of delta-11-desaturase on Bmori_chr.12 [37]. Thus, in our previous cross, we have probably mapped a *trans*-acting regulation of delta-11-desaturase, which may also partly explain the marginal significant effect in BC23.

Variation in the *desat1* desaturase gene of *D. melanogaster* has pleiotropic effects, causing variation in the male and female CHC profiles as well as variation in male response to these profiles [5,6,21]. However, we did not find a pleiotropic effect of the non-functional delta-11-desaturase allele in the male response: homozygous mutant males displayed no difference in physiological or behavioural response to the female sex pheromone. These results are consistent with previous findings that in *H. virescens*, independent and unlinked genes govern the variation in the female pheromone signal and the male pheromone response [17]. This stands in contrast to recent findings in crickets [47], where common genomic regions were found that influence both signal and response, and in *Drosophila*, where a single-linked region was shown to affect both signal and response [7]. We also did not see a pleiotropic effect on larval growth rate or adult mating success. Thus, this desaturase appears to function mainly in the pheromone biosynthetic pathway, as also suggested by its high expression in the sex pheromone gland [37].

A change in female pheromone production followed by a change in male response was previously observed in a laboratory colony of the cabbage looper moth, *Trichoplusia ni*. Crossing experiments showed that a single recessive allele accounted for a rare phenotype showing a 30-fold decrease in (Z)-5-dodecenyl acetate and a 20-fold increase in (Z)-9-tetradecenyl acetate as well as a threefold decrease in the major component (Z)-7-dodecenyl acetate [48], although the gene responsible was never identified. After 50 generations, males in the same laboratory colony changed to be equally responsive to the mutant and wild-type blends, and this behavioural change was correlated with a reduced peripheral sensitivity to the increased component [49]. However, this heritable change in the males was not correlated genetically with the change in female pheromone production.

The large percentage of 16:Ald/Z11–16:Ald variation explained by the premature stop codon in the delta-11-desaturase is quite remarkable. We know of only one other example where such a large proportion of the variance was explained by a single nucleotide polymorphism in a sexually selected trait, i.e. Relaxin-like receptor 2 (RXFP2) that explained up to 76% of the variation in horn size in wild Soay sheep [22]. In this latter case, none of the SNPs would be predicted to inactivate the protein, and a polymorphism is maintained by selective trade-offs in life-history traits [50].

To our knowledge, this is the first time that a single point mutation has been identified that underlies the phenotypic variation in the pheromone signal of a moth. In the fatty-acyl reductase gene responsible for pheromone strain variation in the European corn borer *O. nubilalis*, a total of 73 polymorphic sites were found, 46 of which were fixed strain differences [18], but the causative mutation(s) have not yet been identified. In *D. melanogaster* 36 SNPs in *desat2* were found to be associated with variance in cuticular hydrocarbons and sexual fitness [51]. Because the premature stop codon allele is rare or absent in field populations, it cannot account for the within-population variation in the sex pheromone of *H. virescens* that we previously found in the field [37]. However, as we found that this single mutation has a profound effect on the female sex pheromone signal without affecting the male pheromonal response, or any other fitness component that we could measure, inactivation of the LPAQ delta-11 desaturase could be responsible for the pheromone blend of *Helicoverpa gelotopoeon*, the only heliothine lacking Z11–16:Ald [52]. This would provide a possible example of a saltational shift in moth pheromone blends [53,54].

Ethics. This research was conducted with invertebrates (lepidopteran insects), which are not considered animals, but which always were handled with care.

Data accessibility. The datasets supporting this article have been uploaded as part of the electronic supplementary material.

Authors' contributions. A.T.G. designed and coordinated the study, participated in data analysis and drafted the manuscript. M.v.W. and G.S. carried out the data analyses and statistical analyses. E.V.-P. conducted the wind

tunnel and EAG experiments. P.K. carried out the molecular laboratory work and carried out all sequence alignments. D.v.V. took care of all the rearings and the selection lines. D.G.H. participated in the design and data analysis and helped draft the manuscript. All authors gave final approval for publication.

Competing interests. We have no competing interests.

Funding. This research was supported in part by the National Science Foundation (award IOS-1456973), The Netherlands Organisation for Scientific Research (NWO-ALW, award 822.01.012 and ALWOP 2015.075), the W.M. Keck Center for Behavioral Biology, and the Max Planck Society.

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
