## [Reviewer comments · Royal Society Open Science]

Review History

RSOS-182050.R0 (Original submission)

Review form: Reviewer 1

Is the manuscript scientifically sound in its present form?

Yes

Are the interpretations and conclusions justified by the results?

Yes

Is the language acceptable?

Yes

Is it clear how to access all supporting data?

Yes

Do you have any ethical concerns with this paper?

No

Have you any concerns about statistical analyses in this paper?

Yes

Recommendation?

Accept with minor revision (please list in comments)

Comments to the Author(s)

I begin by reiterating my previous assessment that this is a very well-constructed manuscript that describes an excellent series of experiments. The core result is the discovery of a stop codon mutation in a desaturase gene that has a profound effect on female pheromone blend but no detectable effect on male response or other fitness components. This will certainly be of interest in the field of pheromone communication and the wider field of signal-response evolution.

My principal concern with the version that I saw previously was a tendency to over-state the significance of the results. This arose largely through implying extension of the observations on this laboratory mutation to variation in natural populations. With some reluctance, the authors have adjusted the MS appropriately, in most places, as well as responding to my other minor comments and those of the second reviewer.

I have a few remaining issues. They are all small wording changes, in a sense, but I think they are significant in terms of getting the message right.

Final sentence of abstract – "... a single point mutation that underlies the phenotypic variation...". This is still inclined to give the wrong impression because 'the phenotypic variation' can easily be read as the variation present in nature. "... a single point mutation that alters the pheromone signal of a moth" would be preferable.

69 – I suggest replacing 'within-population variation' with 'difference between lines'. Then on line 73, make clear that this is 99% of the variance in the QTL cross, not in the source population and on line 75 replace 'contributes to the within-population female pheromone variation' with 'can contribute to female pheromone variation'.

258-260 – is 'X-square' here the same as 'Chi-square'? If so, the numbers look to be incorrect. If not, please explain the test.

I don't agree that $p=0.066$ is 'marginal significant' (now line 318), even if the authors 'prefer' this wording (see response). The authors actually describe it as 'just missed significance' on line 225. If the cut-off is $p=0.05$, this is simply not significant.

331-2 – "These results confirm that in *Heliothis virescens*, independent and unlinked genes govern the variation in the female pheromone signal and the male pheromone response." I disagree. The results reveal a single mutation in a single gene that influences pheromone blend but not male response. This may tend to support a general picture of independent genetic determination but it does not 'confirm' it.

339-348 – this is a useful addition. I think that the genetic independence of pheromone blend and male response in *Ostrinia* should also be mentioned at this point. It would also be good to mention here the evidence in *Heliconius* (Lepidoptera) for pleiotropic effects on preference of major genes influencing visual signals. [cited as ref. 24]

In response to Rev. 2, the authors say: "Loss of a functional delta-11 desaturase activity in the female pheromone gland of *H. gelotopoeon* is the only possible explanation of this phenomenon. Therefore it is appropriate to retain the reference to saltational evolution at this point of the Discussion."

The implication is that loss of function must be saltational (now line 370). I don't agree. The end result of loss of function can be reached in a series of small steps.

Review form: Reviewer 2 (Teun Dekker)

Is the manuscript scientifically sound in its present form?

Yes

Are the interpretations and conclusions justified by the results?

Yes

Is the language acceptable?

Yes

Is it clear how to access all supporting data?

Yes

Do you have any ethical concerns with this paper?

No

Have you any concerns about statistical analyses in this paper?

No

Recommendation?

Accept as is

Comments to the Author(s)

The authors have in my opinion responded well to the critique of the reviewers, save some little misunderstanding. I attach a file with some responses and minor points. Nice if the authors can accommodate these (Appendix A).

Best, Teun Dekker

Decision letter (RSOS-182050.R0)

31-Jan-2019

Dear Dr Groot

On behalf of the Editors, I am pleased to inform you that your Manuscript RSOS-182050 entitled "Within-population variability in a moth sex pheromone blend, part 2: Selection towards fixation" has been accepted for publication in Royal Society Open Science subject to minor revision in accordance with the referee suggestions. Please find the referees' comments at the end of this email.

The reviewers and handling editors have recommended publication, but also suggest some minor revisions to your manuscript. Therefore, I invite you to respond to the comments and revise your manuscript.

• Ethics statement

If your study uses humans or animals please include details of the ethical approval received, including the name of the committee that granted approval. For human studies please also detail

whether informed consent was obtained. For field studies on animals please include details of all permissions, licences and/or approvals granted to carry out the fieldwork.

- Data accessibility

If you wish to submit your supporting data or code to Dryad (<http://datadryad.org/>), or modify your current submission to dryad, please use the following link:
<http://datadryad.org/submit?journalID=RSOS&manu=RSOS-182050>

- Competing interests

- Authors' contributions

- Acknowledgements

- Funding statement

Because the schedule for publication is very tight, it is a condition of publication that you submit

the revised version of your manuscript before 09-Feb-2019. Please note that the revision deadline will expire at 00.00am on this date. If you do not think you will be able to meet this date please let me know immediately.

If your manuscript is newly submitted and subsequently accepted for publication, you will be

asked to pay the article processing charge, unless you request a waiver and this is approved by Royal Society Publishing. You can find out more about the charges at <http://rsos.royalsocietypublishing.org/page/charges>. Should you have any queries, please contact openscience@royalsociety.org.

on behalf of Professor Kevin Padian (Subject Editor)
openscience@royalsociety.org

Associate Editor Comments to Author:

The reviewers are largely satisfied with your changes, but recommend some additional modifications to hone an otherwise promising piece of work - good luck and we look forward to receiving the revision.

Reviewer comments to Author:

Reviewer: 1

Comments to the Author(s)

I begin by reiterating my previous assessment that this is a very well-constructed manuscript that describes an excellent series of experiments. The core result is the discovery of a stop codon mutation in a desaturase gene that has a profound effect on female pheromone blend but no detectable effect on male response or other fitness components. This will certainly be of interest in the field of pheromone communication and the wider field of signal-response evolution.

My principal concern with the version that I saw previously was a tendency to over-state the significance of the results. This arose largely through implying extension of the observations on this laboratory mutation to variation in natural populations. With some reluctance, the authors have adjusted the MS appropriately, in most places, as well as responding to my other minor comments and those of the second reviewer.

I have a few remaining issues. They are all small wording changes, in a sense, but I think they are significant in terms of getting the message right.

Final sentence of abstract – "... a single point mutation that underlies the phenotypic variation...". This is still inclined to give the wrong impression because 'the phenotypic variation' can easily be read as the variation present in nature. "... a single point mutation that alters the pheromone signal of a moth" would be preferable.

69 – I suggest replacing 'within-population variation' with 'difference between lines'. Then on line 73, make clear that this is 99% of the variance in the QTL cross, not in the source population and on line 75 replace 'contributes to the within-population female pheromone variation' with 'can contribute to female pheromone variation'.

258-260 – is 'X-square' here the same as 'Chi-square'? If so, the numbers look to be incorrect. If not, please explain the test.

I don't agree that $p=0.066$ is 'marginal significant' (now line 318), even if the authors 'prefer' this wording (see response). The authors actually describe it as 'just missed significance' on line 225. If the cut-off is $p=0.05$, this is simply not significant.

331-2 – “These results confirm that in *Heliothis virescens*, independent and unlinked genes govern the variation in the female pheromone signal and the male pheromone response.” I disagree. The results reveal a single mutation in a single gene that influences pheromone blend but not male response. This may tend to support a general picture of independent genetic determination but it does not ‘confirm’ it.

339-348 – this is a useful addition. I think that the genetic independence of pheromone blend and male response in *Ostrinia* should also be mentioned at this point. It would also be good to mention here the evidence in *Heliconius* (Lepidoptera) for pleiotropic effects on preference of major genes influencing visual signals. [cited as ref. 24]

In response to Rev. 2, the authors say: “Loss of a functional delta-11 desaturase activity in the female pheromone gland of *H. gelotopoeon* is the only possible explanation of this phenomenon. Therefore it is appropriate to retain the reference to saltational evolution at this point of the Discussion.”

The implication is that loss of function must be saltational (now line 370). I don’t agree. The end result of loss of function can be reached in a series of small steps.

Reviewer: 2

Comments to the Author(s)

The authors have in my opinion responded well to the critique of the reviewers, save some little misunderstanding. I attach a file with some responses and minor points. Nice if the authors can accommodate these.

Best, Teun Dekker

Author's Response to Decision Letter for (RSOS-182050.R0)

See Appendix B.

Decision letter (RSOS-182050.R1)

18-Feb-2019

Dear Dr Groot,

I am pleased to inform you that your manuscript entitled "Within-population variability in a moth sex pheromone blend, part 2:

Selection towards fixation" is now accepted for publication in Royal Society Open Science.

Royal Society Open Science operates under a continuous publication model (<http://bit.ly/cpFAQ>). Your article will be published straight into the next open issue and this

will be the final version of the paper. As such, it can be cited immediately by other researchers. As the issue version of your paper will be the only version to be published I would advise you to check your proofs thoroughly as changes cannot be made once the paper is published.

on behalf of Prof Kevin Padian (Subject Editor)
openscience@royalsociety.org

Appendix A

- dont call the components pheromones if there is no evidence of 16Ald having any effect on the behavioral response of males. Call them instead pheromone gland components or something similar. There are several locations (eg line 53, 184), which i leave to the authors to identify and correct.

Answer: In the pheromone literature, there is general agreement on calling a gland chemical a

pheromone component when an attractive response of conspecific males has been shown, and

a pheromone compound when this is not so clear. This is thus the nomenclature that we are

using throughout this manuscript.

I suggest to change pheromone compound to pheromone GLAND compound/component. The difference between component/compound is not so generally understood. I would strongly advise to revise so no-one gets confused.

- line 27 : the dynamic mechanisms. I am not sure if one should call the mechanisms dynamic or rather diverse, the first implying that the mechanisms (eg selected genes that induce change) are themselves dynamic .. well you get the idea

Answer: We don't understand the reviewer's point here, but we have removed the word

"dynamic".

to clarify: dynamic mechanisms imply that the mechanisms themselves are dynamic instead of (what is meant) that mechanisms cause pheromone communication to be dynamic (as implied by evolution).

- line 298: does amount of precursor being generated (in this case 16Ald) depend on the activity of the enzyme that converts it to the pheromone (in this case z11-16Ald)? This is the case in many microbes. I mention this as in the low lines increasing the amount of Z11-16Ald (eg higher enzymatic conversion) could simply lead to more 16Ald be produced (maybe with pleiotropic effects on Z9-14Ald, see also fig 1a and comments below) and thus not change the ratio. In that light it could be good if the authors not only present relative amounts, but also absolute quantities (if the the above would be true one would expect more pheromone compounds and precursors in the low lines)

Answer: We do not understand the reviewer's point, as he writes that an increasing amount of Z11-16:Ald can lead to more 16:Ald, which is opposite to our observations. Moreover, we have not assayed enzyme activity directly. It is the ratio between 16:Ald and Z11-16:Ald that is changed, as is written

throughout the manuscript (i.e the ratio being < 1 or > 1), meaning that more 16:Ald is directly related to less Z11-16:Ald.

To explain: In biological systems, depleting a substrate by higher enzymatic activity can lead to an increased production of the substrate. In this case, when selecting for lower 16:Ald, the selection lines may have favored increased enzymatic activity in the gland. This may have caused more 16:ald being produced (and converted into pheromone components). In effect there is no change in ratio. However, there may be a change in absolute amount of pheromone components being produced.

However, the authors did not mention that the absolute quantity in the low lines was lower than the high lines, which shows that the above is not the case. Please insert a line in the results on the absolute quantity

Appendix B

14 February 2019

Dear editors,

Thank you for your positive decision, we are very pleased that our manuscript is accepted for publication in Royal Society Open Science. Please find our specific answers to the last comments of the reviewer below. We hope that our manuscript is now ready for publication in Royal Society Open Science.

Kind regards, Astrid Groot

Reviewer: 1

I have a few remaining issues. They are all small wording changes, in a sense, but I think they are significant in terms of getting the message right.

Final sentence of abstract - "... a single point mutation that underlies the phenotypic variation...". This is still inclined to give the wrong impression because 'the phenotypic variation' can easily be read as the variation present in nature. "... a single point mutation that alters the pheromone signal of a moth" would be preferable.

Answer: Throughout the abstract we write that this work is done on selection lines. We prefer to keep the wording as is.

69 - I suggest replacing 'within-population variation' with 'difference between lines'.

Answer: We prefer our wording.

Then on line 73, make clear that this is 99% of the variance in the QTL cross, not in the source population

Answer: We don't write 'the source population', we write: "99% of the variance in the pheromone ratio produced by females." In line 69 we specify that this QTL analysis was done on the selection lines.

and on line 75 replace 'contributes to the within-population female pheromone variation' with 'can contribute to female pheromone variation'.

Answer: We prefer our wording. "Can" suggests that there is another possibility, which is not the case in our data.

258-260 - is 'X-square' here the same as 'Chi-square'? If so, the numbers look to be incorrect. If not, please explain the test.

Answer: We thank the reviewer for finding these mistakes, the numbers were indeed incorrect and are now corrected. To omit any possible mistake in writing Chi-square (should be Chi-squared), we now changed this to X^2 .

I don't agree that $p=0.066$ is 'marginal significant' (now line 318), even if the authors 'prefer' this wording (see response). The authors actually describe it as 'just missed significance' on line 225. If the cut-off is $p=0.05$, this is simply not significant.

Answer: we do not agree with the reviewer, the cut-off of a significance of 0.05 is a general agreement, but to dismiss data that are just above this line would be dismissing data that are potentially interesting. Therefore, we point out this result.

*331-2 - "These results confirm that in *Heliothis virescens*, independent and unlinked genes govern the variation in the female pheromone signal and the male pheromone response." I disagree. The results reveal a single mutation in a single gene that influences pheromone blend but not male response. This may tend to support a general picture of independent genetic determination but it does not 'confirm' it.*

Answer: We have changed the wording to read "These results are consistent with previous findings that in *Heliothis virescens*, ..."

*339-348 - this is a useful addition. I think that the genetic independence of pheromone blend and male response in *Ostrinia* should also be mentioned at this point. It would also be good to mention here the evidence in *Heliconius* (Lepidoptera) for pleiotropic effects on preference of major genes influencing visual signals. [cited as ref. 24]*

Answer: To add this example would require another paragraph of explaining that system.

*In response to Rev. 2, the authors say: "Loss of a functional delta-11 desaturase activity in the female pheromone gland of *H. gelotopoeon* is the only possible explanation of this phenomenon. Therefore it is appropriate to retain the reference to saltational evolution at this point of the Discussion." The implication is that loss of function must be saltational (now line 370). I don't agree. The end result of loss of function can be reached in a series of small steps.*

Answer: The reviewer is not correct. We are not implying that loss of function MUST be saltational, as the reviewer claims, but that it MAY be. Our text reads: "This would provide a possible example of a saltational shift..." and we stand by this wording.

=====

Answers to reviewer 2

the behavioral response of males. Call them instead pheromone gland components or something similar. There are several locations (eg line 53, 184), which i leave to the authors to identify and correct.

Answer: In the pheromone literature, there is general agreement on calling a gland chemical a pheromone component when an attractive response of conspecific males has been shown, and a pheromone compound when this is not so clear. This is thus the nomenclature that we are using throughout this manuscript.

I suggest to change pheromone compound to pheromone GLAND compound/component. The difference between component/compound is not so generally understood. I would strongly advise to revise so no-one gets confused.

Answer (10 Feb): We find it important to stick to the nomenclature that is used in the pheromone literature. As explained, pheromone compounds is the generally used term (see e.g. Cardé R & R, Millar JG (eds) 2004. Advances in insect chemical ecology. Cambridge University Press, Cambridge; Wyatt 2010. J. Comp. Physiol. A 196: 685-700). In addition, we have shown that all pheromone compounds present in the gland of *Heliothis virescens* are also emitted (see Lievers R & Groot AT. Plos One *Plos One* 11(8): e0161138). We now added one line in the introduction with this information and the reference (lines 66-67)

- line 27 : the dynamic mechanisms. I am not sure if one should call the mechanisms dynamic or rather diverse, the first implying that the mechanisms (eg selected genes that induce change) are themselves dynamic .. well you get the idea

Answer: We don't understand the reviewer's point here, but we have removed the word "dynamic".

to clarify: dynamic mechanisms imply that the mechanisms themselves are dynamic instead of (what is meant) that mechanisms cause pheromone communication to be dynamic (as implied by evolution).

Answer (10 Feb): Thank you for this clarification. As we removed the word 'dynamic', we assume this point has been resolved.

- line 298: does amount of precursor being generated (in this case 16Ald) depend on the activity of the enzyme that converts it to the pheromone (in this case z11-16Ald)? This is the case in many microbes. I mention this as in the low lines increasing the amount of Z11-16Ald (eg higher enzymatic conversion) could simply lead to more 16Ald be produced (maybe with pleiotropic effects on Z9-14Ald, see also fig 1a and comments below) and thus not change the ratio. In that light it could be good if the authors not only present relative amounts, but also

absolute quantities (if the the above would be true one would expect more pheromone compounds and precursors in the low lines)

Answer: We do not understand the reviewer's point, as he writes that an increasing amount

of Z11-16:Ald can lead to more 16:Ald, which is opposite to our observations.

Moreover, we

have not assayed enzyme activity directly. It is the ratio between 16:Ald and Z11-16:Ald that

is changed, as is written throughout the manuscript (i.e the ratio being < 1 or > 1), meaning

that more 16:Ald is directly related to less Z11-16:Ald.

To explain: In biological systems, depleting a substrate by higher enzymatic activity can lead

to an increased production of the substrate. In this case, when selecting for lower 16:Ald, the

selection lines may have favored increased enzymatic activity in the gland. This may have

caused more 16:ald being produced (nd converted into pheromone components). In effect

there is no change in ratio. However, there may be a change in absolute amount of pheromone

components being produced.

However, the authors did now mentioned that the absolute quantity in the low lines was lower

than the high lines, which shows that the above is not the case. Please insert a line in the

results on the absolute quantity

Answer (10 Feb): We now added the absolute quantities in the results (lines 218-219)